# Influence of the Composition and Vacancy Concentration on Cluster Decomposition Behavior in Al–Si–Mg Alloy: A Kinetic Monte Carlo Study

**DOI:** 10.3390/ma15196552

**Published:** 2022-09-21

**Authors:** Sangjun Lee, Heon Kang, Jonggyu Jeon, Donghyun Bae

**Affiliations:** 1Department of Materials Science and Engineering, Yonsei University, Seoul 03722, Korea; 2Heat & Surface Technology R&D Department, Korea Institute of Industrial Technology, Siheung 15014, Korea

**Keywords:** aluminum alloy, Kinetic Monte Carlo, aging, cluster decomposition, vacancy

## Abstract

The influence of cluster composition and the addition of vacancies on the decomposition behavior of clusters during artificial aging in Al–Si–Mg alloys were analyzed according to the kinetic Montel Carlo model. Clusters with a balanced composition (Mg/(Mg + Si) = 0.5) were the most difficult to decompose. In addition, the cluster decomposition was slower when more vacancies were added to the cluster. Among Si, Mg, and vacancies, vacancies most significantly affect decomposition. The clusters with Mg/(Mg + Si) ≤ 0.4 strongly trap vacancies, which can be classified as hardly decomposable vacancy-rich clusters. The clustering behavior during natural aging and the effect of pre-aging were analyzed using the Kinetic Monte Carlo model. Pre-aging slows down cluster formation due to the lowered vacancy concentration. In addition, the overall composition of the clusters changes to easily decomposable clusters after pre-aging. Thus, not only is the number of clusters reduced but also the clusters are more easily decomposable when pre-aging is performed.

## 1. Introduction

The 6xxx series of Al alloys (Al–Si–Mg-based) are widely used for their good combination of medium strength, formability, and corrosion resistance [1,2]. Si and Mg play roles as solutes, which result in solid solution strengthening, and as sources of precipitates. Precipitate strengthening is the most significant strengthening mechanism of Al–Si–Mg-based alloys [2,3,4,5]. 6xxx series Al alloy sheets go through paint baking during the final manufacturing process, and the range of temperatures is similar to that in which precipitate formation sufficiently occurs [6,7,8]. Therefore, increasing the bake-hardening response (BHR) (that is, the increase in yield strength during paint baking) is one of the main issues in the industrial field [8,9,10].

Solute atoms in the 6xxx series of Al alloys form clusters even at room temperature—a phenomenon known as natural aging (NA) [11,12,13,14]. Vacancies move in the Al matrix according to their activation energy, which involves the exchange of positions with neighboring atoms, a phenomenon known as vacancy diffusion [14,15,16,17]. For precipitate formation, the solution-treated alloy is quenched, and at this time, not only the solutes but also the vacancies are supersaturated. Clusters formed during NA cause non-controllable changes in the mechanical properties [13,18] and decrease the BHR [10,19]. Pre-aging (PA) (short-term artificial aging (AA) performed immediately after solution treatment) is a good way to delay NA and is widely performed in the Al industry [8,10,20,21], wherein the vacancies are rapidly annihilated [22,23]. 

In the early stage of AA, clusters formed during NA can be decomposed, accompanied by a decrease in the yield strength [24,25,26,27]. Subsequently, the solute atoms from the clusters recombine to form β″ precipitates. With the proper decomposition of clusters, the number of solute atoms that can be the source of precipitate formation increases, resulting in a high BHR. The decomposition behavior of clusters depends on their size and composition, and understanding this behavior is essential for designing high-strength alloys.

Previous studies have suggested that Al–Si–Mg alloys with relatively high Si content are vulnerable to NA because Si-rich clusters are difficult to decompose [25,26,27,28]. However, these studies only inferred the decomposition behavior through the mechanical or thermal properties [26,27]. The size of the clusters formed in NA is so small [18] that the clusters are difficult to observe. Some researchers have attempted to analyze the clusters by atom probe tomography [2,28,29]; however, there are still limitations in analyzing detailed mechanisms because of the difficulties in the precise control of early NA stages. 

The structure of the clusters changes even during specimen preparation or observation, considering that clustering occurs at room temperature. Thus, the accurate atomic-scale mechanism of cluster decomposition has not yet been established. Simulation analysis can be a good alternative in solving practical problems. Although several studies have attempted to describe cluster dynamics via *ab-initio* calculations [30,31], the mechanisms for Mg–Si clustering formed during NA in Al–Si–Mg alloys have not been extensively studied. 

The Kinetic Monte Carlo (KMC) model, one of the most fundamental interpretations of phase transformations, is used to describe the motion of atoms at the atomic level [32]. For Al–Si–Mg alloys, simulation of the movement of the solute atoms according to the energy barrier corresponding to the face centered cubic (FCC) lattice is assumed [32]. Although the KMC model has been limited by its excessively high computational cost, it is useful when performing atom-scale analysis that are difficult to conduct experimentally [33].

Thus, the scientific goal here was to reveal the atomic-scale mechanism of cluster formation during NA and cluster decomposition during subsequent AA using the KMC model. Clusters with various compositions and vacancy concentration were presumed, and their migration by vacancy diffusion was described at room temperature (293 K) or under typical AA conditions. Quantitative analysis of the clusters formed during NA was also performed. In addition, NA behavior with/without PA and the cluster decomposition behavior in those cases were also analyzed as a case study for practical thermal process in industrial fields. Understanding the detailed mechanisms of these behaviors is important for designing heat-treatable alloys, thereby, enabling academic and industrial contributions.

## 2. Kinetic Monte Carlo Simulation

### 2.1. Vacancy Diffusion in Al–Si–Mg Alloys

Cluster formation during NA of Al–Si–Mg alloys occurs due to vacancy diffusion caused by excessive vacancy concentration. In the Al lattice, the vacancy migrates in the direction of a nearest neighbor atom with a specific jump frequency, which allows clustering to occur. The larger the frequency and the higher the concentration of vacancies are, the greater the movement of vacancies, and, as a result, clustering occurs more quickly. The vacancy concentration $C_{V}$ is expressed as follows [22,34]:(1)$$C_{V}=\exp\left(-\frac{G_{V}}{k_{B}T}\right)$$
where $k_{B}$ is the Boltzmann constant of 1.38 × 10^23^ J/K and T is the absolute temperature. $G_{V}$ is defined as $G_{V}=H_{V}-TS_{V}$, with $H_{V}$ = 0.67 eV and $S_{V}$ = 0.7 $k_{B}$ [34] taken as values. Al–Si–Mg alloys are generally solution-treated for artificial aging, and since the vacancy concentration is a function of temperature, its value exponentially increases with the solution treatment temperature. For instance, after quenching from 813 K, the vacancy concentration is 2.4 × 10^7^ times higher than the equilibrium value at room temperature (293 K), and the clustering reaction actively occurs even at room temperature. 

After solution treatment, some of the Al substitutional sites in the FCC lattice are replaced with solute atoms of Si and Mg depending on the chemical composition of the alloy. Similarly, some of the Al sites are replaced with vacancies depending on the concentration. In the FCC lattice, the nearest neighbor sites from an atom are along 12 directions, and the vacancy jump frequency toward site *i* can be expressed as follows [32]:(2)$$w_{i}=v_{i}\exp\left(-\frac{E_{i}^{a}}{k_{B}T}\right)\;\left(i=1,2,3,\cdots ,12\right)$$
where $v_{i}$ is coefficient for the jump attempt frequency of atom i and $E_{i}^{a}$ is the activation energy of a vacancy jumping toward atom i. $E_{i}^{a}$ is the sum of the differences in the diffusion activation energy for the diffusion of an element and vacancy generation at site i as well as the migration energy toward site i for vacancies. Therefore, the activation energy of jumping from site j to site i is expressed as follows [32]:(3)$$E_{i}^{a}=E_{i}-E_{V}-\left(\underset{k\in NN_{j}}{\sum }\varepsilon_{k\text{–}V_{j}}+\underset{k\in NN_{i}}{\sum }\varepsilon_{k\text{–}i_{i}}\right)+\left(\underset{k\in NN_{i}}{\sum }\varepsilon_{k\text{–}V_{i}}+\underset{k\in NN_{j}}{\sum }\varepsilon_{k\text{–}i_{j}}\right)$$
where $E_{i}$ is the diffusion activation energy of atom i, $E_{V}$ is the vacancy formation energy, $\varepsilon_{k\text{–}i}$ is the chemical interaction energy of atom k and one of its nearest neighbor atoms i, and $\varepsilon_{k\text{–}V}$ is the chemical interaction energy of atom k and a vacancy. $NN_{i}$ denotes the nearest neighbor of site i (12 sites in this case), and k denotes the specific position among the nearest atoms. The parameters required for the above equations are listed in Table 1. The jump attempt frequency coefficient is a pre-factor obeying Arrhenius equation and is independent to temperature [35]. 

The activation energy for Al/solutes is also maintained with respect to temperature. Mantina et al. suggested that correlation factors for the activation energy calculation compensate the difference from varying temperature [35]. The chemical interaction energy is typically used to describe NA and AA behaviors in Al–Si–Mg alloys [36,37,38]. Therefore, three types of used parameters are valid under the given temperature conditions. The chemical interaction energies of the Al–solute/vacancy and vacancy–vacancy were assumed to be zero [39]. The sign of the parameters related to energy denotes the direction of the interaction (+ repulsive and − attractive).

**Table 1 materials-15-06552-t001:** Values of the parameters used in Equations (2) and (3).

Parameter	Value	Ref.
$v_{Al}$ (s^−1^)	1.66 × 10^13^	[35]
$v_{Mg}$ (s^−1^)	1.86 × 10^13^	[35]
$v_{Si}$ (s^−1^)	1.57 × 10^13^	[35]
$E_{Al}$ (eV)	1.29	[35]
$E_{Mg}$ (eV)	1.27	[35]
$E_{Si}$ (eV)	1.15	[35]
$E_{V}$ (eV)	0.63	[35]
$\varepsilon_{Mg\text{–}V}$ (eV)	−0.015	[39]
$\varepsilon_{Si\text{–}V}$ (eV)	−0.025	[39]
$\varepsilon_{Mg\text{–}Si}$ (eV)	−0.04	[39]
$\varepsilon_{Mg\text{–}Mg}$ (eV)	0.04	[39]
$\varepsilon_{Si\text{–}Si}$ (eV)	0.03	[39]

Considering the simulation of the clustering reaction at the atomic level, multiple Al FCC unit cells are set as an area, and this simulation area is assumed to be periodic at all faces of the area. Since the room temperature at which NA occurs is not a sufficient condition to change the crystal structure, the solute atoms and vacancies can be assumed to be located at the sites of FCC Al [32]. 

Therefore, multiple sites of some Al atoms in the simulation area are replaced with Si and Mg according to the chemical composition of the alloy (assuming that the alloy is fully solid-solutioned), and one site is replaced with a vacancy. The replacement position is randomly determined. For each of the 12 nearest neighbor atoms of a vacancy, the jump frequency $w_{i}$ can be obtained, and each value is related to the probability of the vacancy jumping in that direction [40]. 

An arbitrary number m between 0 and 1 is randomly taken for every simulation loop, and the vacancy changes its position with the nearest atom at site k under the condition $\sum_{i=1}^{k-1}w_{i}<m\sum_{i=1}^{12}w_{i}\leq \sum_{i=1}^{k}w_{i}$. The simulation loop repeats until the desired number of times. The schematic representation of the simulation methods by the KMC model is shown in Figure 1.

### 2.2. Modeling Methods

#### 2.2.1. Cluster Decomposition Behavior

To analyze the decomposition behavior of clusters with various compositions, 60 solutes of clusters with different Mg/(Mg + Si) ratios were considered in the Al matrix at a temperature of 453 K, which is a typical condition for bake hardening in Al–Si–Mg-based alloys [8,41]. The size of the Al matrix was set to 20 × 20 × 20 FCC unit cells, a side length of which (~8.9 nm) is smaller than the typical inter-dislocation spacing in Al–Si–Mg alloys (<10 nm [42,43]). As NA results from vacancy diffusion, the inter-dislocation spacing should be considered to avoid the effect of dislocations at which vacancies are annihilated [22,44]. 

Solute clusters typically contain substantial concentration of vacancies after long-term NA [28]. Thus, one, two, or three vacancies were also added to reveal the influence of vacancies in cluster decomposition. Spherical morphology was assumed for the cluster, and the configurations of Si and Mg were randomly selected. 

After constructing the clusters, a vacancy was added outside of the cluster, corresponding to the vacancy existing on the Al matrix. For example, there are two vacancies in the simulation box containing Mg–Si cluster with one vacancy. The simulation results were averaged after more than 30 trials. The decomposition rate is expressed as Monte Carlo steps (MCS). In a simulation box containing more than one vacancy, MCS is counted every single time a vacancy diffuses, and each vacancy is presumed to diffuse alternately. 

This is because the effect of vacancy concentration in the overall system should be compensated by counting MCS more. In addition, to analyze the trapping behavior of vacancies inside clusters during NA with various compositions, clusters composed of 60 solutes and single vacancies with various Mg/(Mg + Si) ratios were considered at 293 K. The degree of trapping was determined as MCS immediately after the vacancy escaped the cluster. The results were averaged after more than 100 trials, and outliers (10 maximum and minimum values) were eliminated.

#### 2.2.2. Clustering during NA with/without PA

To address an example of the industrial thermal process through the conclusions we obtained (the details will be shown in Section 3), NA behaviors with/without PA at 423 K for 10 min in an Al–Si–Mg alloy were also investigated. Considering the clustering reaction in NA, an Al FCC unit cell with a size of 20 × 20 × 20 was set as the simulation area, and 505 positions of Al were replaced with 363 Si and 142 Mg atoms (Al–1.13 Si–0.44 Mg (at.%)). The given alloy was assumed to be quenched after solution treatment at 813 K—a typical condition for solution treatment in Al–Si–Mg alloys [45,46]. 

Atoms existing on the six boundary faces of the cube-shaped simulation area may escape the area with a non-zero probability. As the overall system was assumed to be periodic, the escaped atoms were allowed to enter the opposite side such that the total number of solute atoms was unchanged. Clusters connected on opposite sides were considered as a single cluster. As the FCC crystal structure is unchanged during room temperature maintenance, several Si and Mg atoms gathered at the nearest positions were defined as a cluster. The KMC simulation to describe clustering during NA with/without PA was conducted according to Section 2.1. After a simulation loop, the next loop proceeds after reflecting the physical time lapse, the value of which can be expressed as follows [32]:(4)$$\Delta t=\frac{1}{N_{V}C_{V}\sum_{i=1}^{12}w_{i}}$$
where $N_{V}$ is the total number of atoms in the simulation area. To simplify the calculation, the simulation is performed under single vacancy conditions; however, this can be compensated by dividing by the $N_{V}C_{V}$ term. The above process is repeated until the selected time t.

## 3. Results and Discussion

Figure 2 shows the number of residual atoms composing the clusters at 453 K with respect to the MCS. The number of residual atoms gradually decreased as MCS increased for every condition, indicating the cluster decomposition. The decrease rate in the number of residual atoms depends on the vacancy concentration and composition of the original cluster (Figure 2). The number of residual atoms most slowly decreased when the composition of the original clusters was balanced, (Mg/(Mg + Si) = 0.5) and rapidly decreased when the cluster composition was biased toward Si-rich or Mg-rich. This observation indicates that decomposition of the clusters with a balanced composition is relatively slow. This relationship also occurred regardless of the addition of vacancies, as shown in Figure 2. 

Furthermore, the decomposition rate decreased as the clusters contained more vacancies for each cluster composition. With respect to the clusters with balanced composition, the numbers of residual atoms after 10^6^ MCS without vacancies and with one, two, and three vacancies were approximately 4.03, 10.33, 12.17, and 20.01, respectively. 

This relationship occurs in other cluster compositions. Therefore, clusters with a biased composition containing few vacancies formed during NA are more actively decomposed in the subsequent AA, and inducing the formation of these types of clusters during NA is advantageous for attaining a high BHR.

To quantify the decomposition rate, the Johnson–Mehl–Avrami equation, which has been widely used to describe the phase transformation behavior, was used and can be expressed as follows [47,48]:(5)$$X=1-\exp\left(-kt\right)$$
where X is the degree of phase transformation expressed as a ratio, k is the rate constant, and t is time. The ratio of the number of residual atoms in the clusters to the total number of atoms was presumed to the degree of cluster decomposition. Figure 3a shows the rate constant k with respect to Mg/(Mg + Si) without vacancies and with the addition of one, two, and three vacancies. The rate constant gradually increased as the Mg/(Mg + Si) ratio increased, whereas it gradually decreased when the Mg/(Mg + Si) ratio was greater than 0.5. 

These trends are maintained with the addition of vacancies. Furthermore, the overall rate constants for the cluster decomposition decrease as more vacancies are added. The results in Figure 2 show that the clusters with balanced compositions and clusters with more vacancies are slowly decomposed, as shown in Figure 3a. 

Figure 3b also illustrates the relationship between the rate constant and cluster composition or vacancy concentration; it demonstrates the rate constant k with respect to the vacancy concentration for various Mg/(Mg + Si) ratios. The rate constant gradually decreased as the vacancy concentration increased for each Mg/(Mg + Si) condition. Additionally, the rate constant was the lowest when the cluster composition was balanced, which gradually increased as the composition became biased. The relationship between the rate constant and cluster composition can be explained by combining the overall interaction energy between solutes. 

Clusters composed of Si and Mg contain three types of bonding: Si–Si, Mg–Mg, and Mg–Si. The two-body interaction energies between Si–Si, Mg–Mg, and Mg–Si are 0.03, 0.04, and −0.04 eV, respectively [39]. The homogeneous interaction energies (Si–Si and Mg–Mg) have positive values, whereas the interaction energy of Mg–Si has a negative value, implying repulsive and attractive interactions, respectively. The two-body interaction energies between Si–vacancy and Mg–vacancy are −0.025 and −0.015 eV, respectively, indicating that the vacancies interact negatively with Si and Mg [39]. 

The overall interaction energy of the cluster is the sum of the interaction energies between neighboring solutes. When the overall interaction energy of the cluster increases, the magnitude of the overall repulsive force inside the cluster increases, resulting in easy decomposition of the cluster. Conversely, the cluster decomposition becomes increasingly difficult as the overall interaction energy of the cluster increases negatively. A cluster with a biased composition contains more homogeneous interactions compared with Mg–Si because a specific solute composes most of the cluster, and the proportion of interactions between identical solutes is the highest. 

However, clusters with balanced compositions contain the most Mg–Si interactions. Table 2 shows the fractions of Si–Si, Mg–Mg, and Mg–Si interactions within the cluster with respect to the Mg/(Mg + Si) ratios. Each fraction was taken from 500 random configurations of Mg–Si clusters with 60 solute atoms. According to the combination of interaction energies composing a cluster, the rate constant of clusters with similar contents of Si and Mg is relatively low; that is, clusters with balanced compositions are difficult to decompose, resulting in detrimental effects on the subsequent AA behavior. The addition of vacancies leads to a similar effect regardless of the composition of clusters as both Si–vacancy and Mg–vacancy interactions negatively contribute to the overall interaction energy of clusters.

Figure 3c shows the change in the average absolute rate of k with Si, Mg, and vacancy concentrations. In the case of Si and Mg, the results where the target solute was more than the others were only considered because the increase or decrease trend transitioned to the Mg/(Mg + Si) ratio of 0.5. Each value represents the negative contribution of each element to the cluster decomposition. The dk/dC values (C denotes the concentration) of Si, Mg, and the vacancies were approximately 0.0339, 0.0369, and 0.01436, respectively. 

Although the dk/dC value of Mg was slightly greater than that of Si, it could be regarded as having almost no dependence. However, the dk/dC value of the vacancies was much greater than those of Si and Mg, indicating that the vacancies dominantly contributed to the inhibition of cluster decomposition during NA. 

For the addition of Si or Mg solutes in an arbitrary cluster composition, the newly added solutes interacted homogeneously and heterogeneously, and each type of interaction countervailed. However, vacancies only interacted negatively with Si or Mg solutes. The attractive force between the elements composing the clusters became much stronger with the addition of vacancies, and the contribution of vacancies to cluster decomposition was more dominant than with the addition of Si or Mg solutes. In particular, to obtain a reasonable BHR, a proper alloy composition and thermomechanical process should be designed such that fewer vacancies are present in the cluster.

Figure 4 shows the vacancy escape times with respect to the Mg/(Mg + Si) ratios. Although the vacancies rapidly escaped from clusters composed of similar Si and Mg contents, they were trapped for a long time when the cluster composition was biased. Remarkably, Si-biased clusters trapped vacancies more strongly than Mg-biased clusters owing to the negatively stronger Si–vacancy interaction compared with that of Mg–vacancy. 

The escape time of the vacancy soared when the Mg/(Mg + Si) ratio was less than 0.4. Aruga et al. previously classified the Si-rich clusters as Mg/(Mg + Si) ≤ 0.4 and concluded that Si-rich clusters do not decompose during subsequent AA or act as precursors of β″ precipitates [28]. According to Figure 4, Si-rich clusters as suggested by Aruga et al., can be regarded as vacancy-rich clusters, considering the importance of vacancy in cluster decomposition. Moreover, considering that vacancies dominantly inhibit cluster decomposition and are strongly trapped in cluster with Mg/(Mg + Si) ≤ 0.4, the calculated results are in good agreement with the experimental results in other studies [25,28].

To illustrate an example of an industrial thermal process, the clustering behavior during NA with/without PA at 423 K for 10 min was analyzed using the KMC model. The cluster decomposition behavior was also analyzed according to the conclusions obtained above. Figure 5 shows the cluster number density, average size, and fraction of solutes participating in the clustering reaction with respect to the NA time when PA was performed at 423 K for 10 min on the investigated alloy. The clustering reaction intuitively occurred faster when only NA was performed; without PA, the cluster number density rapidly increased until 30 min and then slowly decreased (Figure 5a). 

The fraction of solutes participating in the clustering reaction also increased until 30 min, after which there was no significant difference; the fraction was saturated with a value of approximately 0.6 (Figure 5c). The cluster number density increased until 30 min of NA because the solutes were adequately supplied to the clustering reaction; in particular, the nuclei of the clusters are continuously generated. Subsequently, no significant difference was observed in the solute amount, and the number density tended to decrease. The average size of the clusters steadily increased with the NA time (Figure 5b). 

In particular, few additional solutes participated in the reaction after 30 min, and the cluster number density decreased as the existing clusters became coarse. In the case of NA after PA, the factors indicating the degree of reaction—that is, the cluster number density, average size, and fraction of solutes participating in the clustering reaction—more slowly increased than in the case of NA without PA (Figure 5). The vacancy diffusion became slow in the investigated alloys with relatively low vacancy concentrations owing to PA [20,49], which can be confirmed according to Equation (4). However, the starting point of the factors indicating the degree of reaction were higher than those without PA owing to clustering during PA at 423 K for 10 min.

To analyze the clusters quantitatively, the size distribution function of the cluster is defined as follows:(6)$$f\left(x\right)=\frac{N_{x}x}{a}$$
where $N_{x}$ is the number of clusters containing x solute atoms and a is the total number of solutes in the simulation area. Figure 6 shows the size distribution function of the clusters for size and composition after 1440 min of NA, 10 min of PA at 423 K, and 1440 min of NA with 10 min of PA at 423 K. The size distribution functions of each case reflect the results shown in Figure 5. Only a limited composition of clusters existed for sufficiently large clusters, whereas the Mg/(Mg + Si) ratios varied more when the cluster size was smaller. 

Remarkably, the average Mg/(Mg + Si) ratio of clusters formed during NA with PA was closer to 0.5 than that without PA. (Mg/(Mg + Si) was already closer to 0.5 only after PA compared to that of NA without PA.) As the average Mg/(Mg + Si) ratio of clusters with less than 10 atoms in Figure 6a was 0.415, which was similar to the overall value, the more-balanced composition of clusters resulted from PA and not the cluster size. 

The interactions between clusters with balanced compositions and vacancies were relatively weak (Figure 4), thereby, resulting in the easy decomposition of the cluster during subsequent AA. In addition, fewer vacancies existed in the given alloy after PA (Equation (1)), resulting in less vacancy content in the clusters formed after PA. Thus, the lowered vacancy concentration during PA decelerates vacancy diffusion and makes the cluster easily decomposable—that is, detrimental effects of NA on subsequent AA behavior are inhibited.

## 4. Conclusions

In summary, the decomposition behavior of clusters formed during NA in subsequent AA differs according to the cluster composition and vacancy concentration. Vacancies dominantly influence the inhibition of cluster decomposition because the addition of vacancies causes the overall interaction energy of the cluster to be negative. In Al–Si–Mg alloys, clusters with a composition of Mg/(Mg + Si) ≤ 0.4 effectively trap vacancies, and thus vacancy-rich clusters can be classified as the criterion. 

After PA, cluster formation during NA was slower than that without PA owing to the low vacancy concentration. The composition of the clusters formed during NA with PA was closer to being balanced—forming clusters after PA that interact weakly with vacancies. Moreover, clusters formed after PA contained fewer vacancies owing to vacancy annihilation during PA. According to the low vacancy content, clusters formed after PA are easily decomposed in the subsequent AA and are advantageous for sufficiently increasing the BHR.

## Figures and Tables

**Figure 1 materials-15-06552-f001:**
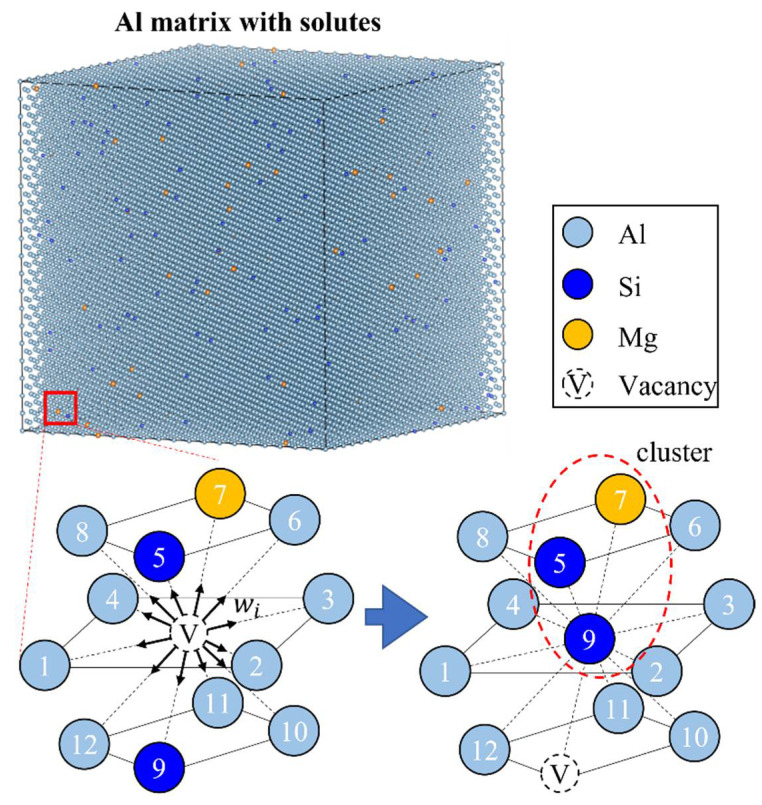
Schematic representation of the clustering mechanism in Al–Si–Mg according to the KMC model.

**Figure 2 materials-15-06552-f002:**
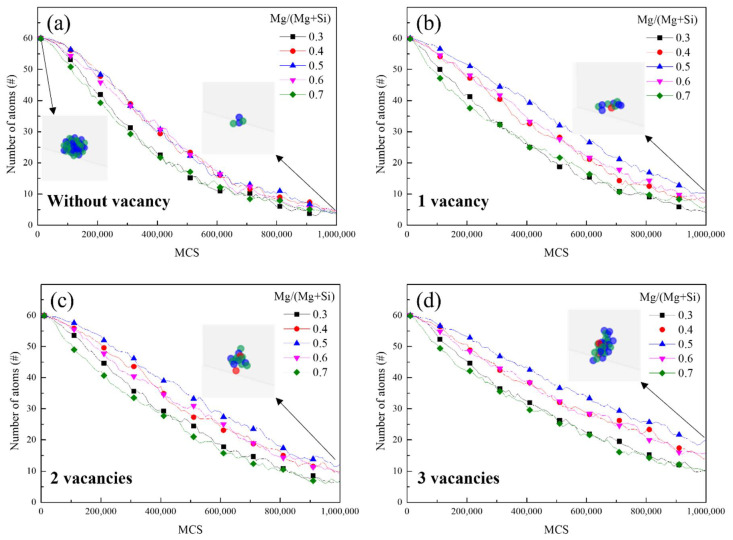
The number of residual atoms composing clusters with respect to the Monte Carlo steps (MCS) for various Mg/(Mg + Si) ratios at 453 K: (**a**) without vacancy, (**b**) with one vacancy, (**c**) with two vacancies, and (**d**) with three vacancies. The black arrows indicate the representative schematic image of clusters for each condition. Green, blue, and red represent Si, Mg, and vacancies, respectively. Al atoms and dissolved solutes are not shown.

**Figure 3 materials-15-06552-f003:**
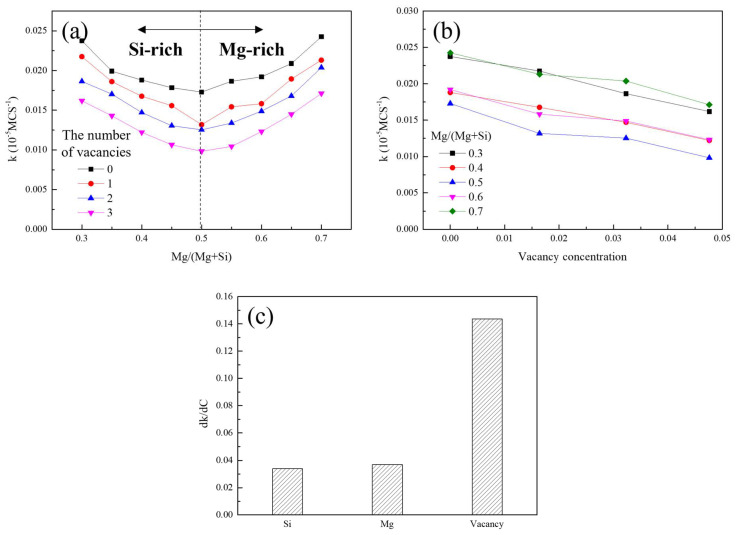
The rate constants of the cluster decomposition with respect to (**a**) Mg/(Mg + Si) ratios for various vacancy conditions and (**b**) vacancy concentration for various Mg/(Mg + Si) ratios. (**c**) Bar graph of average rate of the k value (rate constant) of cluster decomposition with Si, Mg, and vacancy concentrations for various Mg/(Mg + Si) ratios or vacancy concentrations.

**Figure 4 materials-15-06552-f004:**
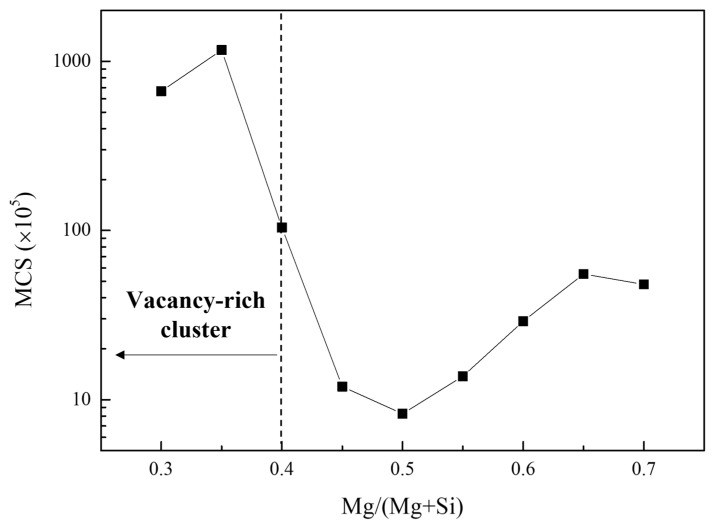
The average Monte Carlo step (MCS) value when a vacancy escaped from the 60 atoms of a cluster at 293 K with respect to the Mg/(Mg + Si) ratio.

**Figure 5 materials-15-06552-f005:**
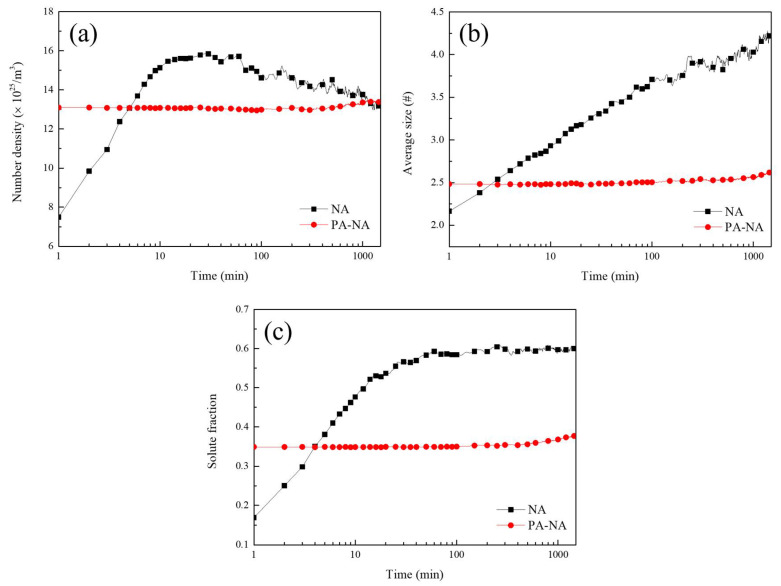
Several properties that represent the degree of the clustering reaction with respect to the natural aging (NA) time when pre-aging (PA) was performed at 423 K for 10 min or not: (**a**) the number density of clusters, (**b**) the average size of clusters in terms of the number of atoms, and (**c**) the fraction of solute atoms participating in the clustering reaction.

**Figure 6 materials-15-06552-f006:**
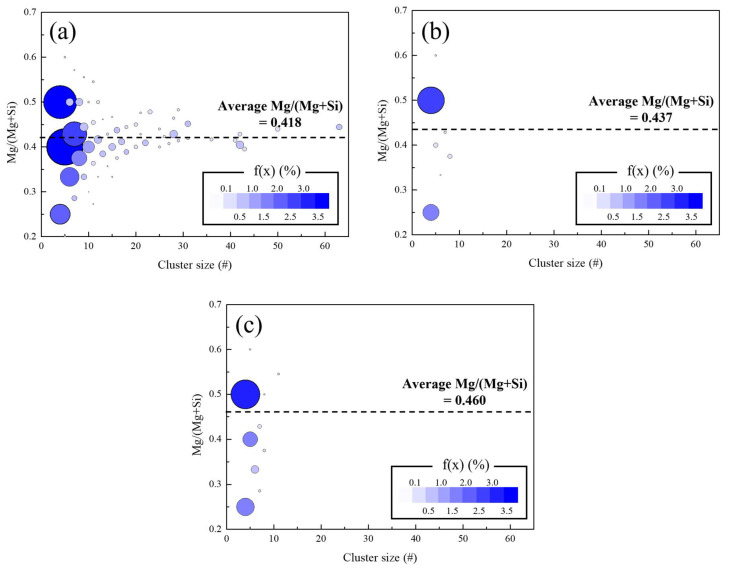
Color maps of the Mg/(Mg + Si) ratio with respect to the size of clusters: (**a**) 1440 min of natural aging (NA), (**b**) 10 min of pre-aging (PA) at 423 K, and (**c**) 1440 min of NA after 10 min of PA at 423 K. The colors and relative sizes denote the size distribution function of clusters with a specific size. For clarity, dimer and trimers are not shown because their size distribution function is much greater than that of the other sizes of clusters.

**Table 2 materials-15-06552-t002:** Fractions of Si–Si, Mg–Mg, and Mg–Si interactions with respect to the Mg/(Mg + Si) ratios.

Mg/(Mg + Si)	0.3	0.35	0.4	0.45	0.5	0.55	0.6	0.65	0.7
Si–Si (%)	48.8	41.93	35.75	29.58	24.42	19.74	15.54	11.95	8.72
Mg–Mg (%)	8.84	11.72	15.47	19.99	24.56	29.93	35.42	41.84	48.68
Mg–Si (%)	42.49	46.34	48.77	50.42	51.02	50.33	49.04	46.21	42.6

## Data Availability

The data underlying this article will be shared upon reasonable request of the corresponding author.

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
