# Peer review of "Influence of the Composition and Vacancy Concentration on Cluster Decomposition Behavior in Al–Si–Mg Alloy: A Kinetic Monte Carlo Study"

_materials, 2022, doi:10.3390/ma15196552_

Round 1
Reviewer 2 Report
Paper is addressing vacancy mediated decomposition in Al alloy treated with kMC simulations. It is interesting and has potential to be published, however there are some issues need to be further discussed prior to that.
1. In kMC simulations of vacancy diffusion exchange rate depends on the temperature. Paper in table 1 list parameters, but temperature for which those are valid is not given.
2. In connection to above there are simulations of pre aging at 423 K prior to aging at 293 K. What energy constants were used for simulations at 423 K or were both performed with the same parameters? If latter is used then simulations are not realistic.
3. Simulation box size used was 20x20x20 FCC lattice constants. Did you performed any simulation to determine size effects since simulation box is quite small and with more than one vacancy in the box could have strange behaviour.
4. When more than one vacancy in box vas used how was rate calculation preformed? Since as with increase of the number of vacancies number of possible events also increases. Moreover, there seems that vacancy-vacancy interaction energy is not listed.
5. In eq. 2 you explain how physical time is calculated, but in all subsequent presentation only MCS is used. Is there any good reason why time is not used.
6. In figure 2a what exactly is meant as "without vacancy"? If there really is simulation without vacancies then kinetics is changed from vacancy diffusion for which you list parameters to direct solute exchange which has totally different parameters.
7. Description of equation 5 in the text is not correct and should be properly described.
8. In lines 206-208 you try to explain that vacancy interact negatively with Si and Mg which is not true. Vacancy trapping depends on cluster composition where in Mg or Si rich clusters vacancy behaviour is still exchanges with neighbours, but since Mg-Mg and Si-Si mixing energies are repulsive it gets trapped inside cluster.
9. In the paper you are stating that neutral cluster composition is Mg/(Mg+Si)=0.5, however, in reality cluster should have composition of Mg2Si, where ratio is Mg/(Mg+Si)=0.666. Please explain this discrepancy.
10. Please use SI units. In line 296 used unit d is not know and should be explained. Also why are results suddenly switched from MCS to physical time?
11. In conclusion you state that clusters were formed with few vacancies, but research was not conducted on number of vacancies formed. Also vacancy interaction weakly or strongly depends on cluster solute composition and not not number of vacancies.
Reviewer 3 Report
This manuscript offers a concise yet compelling report on the influence of the composition and vacancy concentration on cluster decomposition behavior in Al–Si–Mg alloy by employing a kinetic Monte Carlo study. The emphasis of the study falls on the clustering behavior during natural aging as well as on the effect of pre-aging. The chosen kinetic Monte Carlo approach and the level of theory in general, but also the discussion provided are quite adequate for the present ambitious purpose. The nice comparative context of the simulation results of the clustering mechanism and throughout the whole study contributes to reliable scrutinization and prompts good understanding of how the composition and vacancy concentration influence the cluster decomposition behavior in Al–Si–Mg alloy – all this with potentially high practical impact.
From practical point of view, the reported results thus bring new knowledge and certainly represent an original contribution in the present context.
The authors chose an adequate structure of the manuscript – an excellent point of departure for such a study. Finally, the authors provided a balanced realistic and nicely illustrated presentation of their results and corresponding analysis that is of much scientific and practical interest and adds new knowledge to the field.
In my opinion, the fine detailing in the present work, the insightful and balanced discussion of the results, as well as the excellent, intuitively perceived figures, permit wide circle of readers to utilize the manuscript as a guidance for their potential future work in the same or in a similar research field. Consequently, this manuscript presents an efficient and beneficial basis for promoting and solving next step challenges in this field.
The manuscript also benefits from a clear motivation, and it is an easy and informative read.
The present manuscript is a significant contribution, this work once published would be quite useful as well as instructive and suggestive in terms of further studies and to a wider readership.
There are some minor issues with this already excellent manuscript that will need to be addressed before becoming suitable for publication, i.e., it can be considered for publication after a minor revision:
1: Title should be corrected “… alloy: a kinetic Monte Carlo study”.
2: In the introduction, the authors miss that there is a wide range of higher level theoretical/simulation results more ambitious levels of theory for studying similar aspects in composition and and clustering (incl compounds containing metals), such results give credibility of further kinetic MC approaches used at larger size scale. Examples include Physical Journal of Physics: Condensed Matter 27 (2015) 485306, Dalton Transactions 44 (2015) 3356-3366; Such works should be referred to.
3: The authors should elaborate and comment on the explicit thermal aspects as related to the industrial thermal process concerning the alloy studied and discussed hereby. Are there any direct limitations related to the thermal process and how this can impact the present discussion?
4: It would be helpful and valuable to the general readership if bonding (especially in terms of bond characteristics) is commented in more quantitative details.
5: Spell-check and stylistic revision of the paper are still necessary. Some, long sentences, misspellings, etc., still are noticeable throughout the text.
Round 2
Reviewer 1 Report
The manuscript has been well revised,thus I agree to publish it
Reviewer 2 Report
Paper is significantly improved, with comments properly defended or rebutted.